# Pancreatic Cancer Treatment Targeting the HGF/c-MET Pathway: The MEK Inhibitor Trametinib

**DOI:** 10.3390/cancers16051056

**Published:** 2024-03-05

**Authors:** Junyeol Kim, Tae Seung Lee, Myeong Hwan Lee, In Rae Cho, Ji Kon Ryu, Yong-Tae Kim, Sang Hyub Lee, Woo Hyun Paik

**Affiliations:** Department of Internal Medicine, Liver Research Institute, Seoul National University Hospital, Seoul National University College of Medicine, Seoul 03080, Republic of Korea; kimjun16@gmail.com (J.K.); rhytksa@snu.ac.kr (T.S.L.); dlaudghks01@snu.ac.kr (M.H.L.); inrae0428@snu.ac.kr (I.R.C.); jkryu@snu.ac.kr (J.K.R.); yongtkim@snu.ac.kr (Y.-T.K.); gidoctor@snu.ac.kr (S.H.L.)

**Keywords:** pancreatic cancer, trametinib, tumor microenvironment

## Abstract

**Simple Summary:**

Pancreatic cancer, notorious for its aggressive nature and complex desmoplastic tumor microenvironment, poses substantial treatment challenges, particularly in terms of early detection, and often exhibits resistance to standard chemotherapy and immunotherapy. Trametinib is a new therapeutic drug, a selective inhibitor of MEK1 and MEK2, which has shown promise in addressing these challenges. It not only hampers cancer cell division, angiogenesis, and metastatic spread but also significantly modifies the tumor microenvironment. This modification includes altering the activities of fibroblasts and immune cells, which are crucial in the cancer’s progression and response to treatments. Several pivotal studies have highlighted trametinib’s efficacy in reducing the proliferation of pancreatic cancer cells and enhancing the effectiveness of combined treatment regimens, especially in cases that show resistance to conventional therapies. The drug’s potential to improve patient outcomes is currently under investigation in various clinical trials. These trials, encompassing different stages and forms of pancreatic cancer, are instrumental in determining the optimal use of trametinib, both as a single agent and in combination with other drugs. The ongoing research is crucial in defining trametinib’s role in the evolving therapeutic landscape for pancreatic cancer, aiming to provide new avenues for treatment and potentially improve survival rates in this challenging disease area.

**Abstract:**

Pancreatic cancer is characterized by fibrosis/desmoplasia in the tumor microenvironment, which is primarily mediated by pancreatic stellate cells and cancer-associated fibroblasts. HGF/c-MET signaling, which is instrumental in embryonic development and wound healing, is also implicated for its mitogenic and motogenic properties. In pancreatic cancer, this pathway, along with its downstream signaling pathways, is associated with disease progression, prognosis, metastasis, chemoresistance, and other tumor-related factors. Other features of the microenvironment in pancreatic cancer with the HGF/c-MET pathway include hypoxia, angiogenesis, metastasis, and the urokinase plasminogen activator positive feed-forward loop. All these attributes critically influence the initiation, progression, and metastasis of pancreatic cancer. Therefore, targeting the HGF/c-MET signaling pathway appears promising for the development of innovative drugs for pancreatic cancer treatment. One of the primary downstream effects of c-MET activation is the MAPK/ERK (Ras, Ras/Raf/MEK/ERK) signaling cascade, and MEK (Mitogen-activated protein kinase kinase) inhibitors have demonstrated therapeutic value in RAS-mutant melanoma and lung cancer. Trametinib is a selective MEK1 and MEK2 inhibitor, and it has evolved as a pivotal therapeutic agent targeting the MAPK/ERK pathway in various malignancies, including BRAF-mutated melanoma, non-small cell lung cancer and thyroid cancer. The drug’s effectiveness increases when combined with agents like BRAF inhibitors. However, resistance remains a challenge, necessitating ongoing research to counteract the resistance mechanisms. This review offers an in-depth exploration of the HGF/c-MET signaling pathway, trametinib’s mechanism, clinical applications, combination strategies, and future directions in the context of pancreatic cancer.

## 1. Pancreatic Cancer and Its Tumor Microenvironment

Pancreatic cancer is an aggressive disease and currently ranks as the third leading cause of cancer-related deaths in the United States, surpassing breast cancer [1]. The majority of cases involve exocrine pancreatic cancers, including pancreatic ductal adenocarcinomas and pancreatic acinar cell carcinomas [2]. Despite advancements in chemotherapy and other treatment modalities, the long-term survival prospects for patients with pancreatic cancer remain dismally low, with the five-year survival rates lingering around 10% [3,4].

The poor prognosis is largely attributed to the fact that most patients are diagnosed at an advanced, metastatic stage, making them not feasible for curative surgical resection. Additionally, a substantial number of patients encounter cancer recurrence even after surgical resection. Moreover, pancreatic cancer is notorious for its resistance to conventional treatment. This resistance is observed across various treatment modalities, including chemotherapy and immunotherapy, such as immune checkpoint-blocking agents like anti-programmed death-1/programmed death-L1 antibodies and anti-cytotoxic T lymphocyte antigen-4 antibodies.

Notably, meaningful treatment responses in pancreatic ductal adenocarcinoma are predominantly limited to a specific subtype of the disease, known as the classical glandular subtype. This subtype is characterized by its epithelial morphology and a gene expression program that reflects a more differentiated state [5,6,7]. In contrast, undifferentiated non-glandular pancreatic ductal adenocarcinoma, which exhibits a mesenchymal morphology and a basal-like transcriptional program, represents a particularly aggressive form [5,6,7,8,9,10]. These tumors are marked by their poor prognosis and a notable lack of responsiveness to standard-of-care chemotherapy. The mesenchymal morphology of these tumors is indicative of a more invasive and metastatic phenotype, which contributes to their aggressive clinical behavior. Additionally, the basal-like transcriptional program of these non-glandular pancreatic ductal adenocarcinomas suggests a fundamental difference in their cellular biology, which may underlie their resistance to therapies, including targeted and immunotherapies [11]. The challenge in treating non-glandular pancreatic ductal adenocarcinoma lies in its intrinsic resistance mechanisms, which are not yet fully understood. This resistance is multifaceted, involving not only the tumor cells themselves but also the tumor microenvironment.

Given these challenges, non-glandular pancreatic ductal adenocarcinomas represent a critical unmet clinical need. Developing effective therapeutic strategies for this subtype of pancreatic ductal adenocarcinoma requires a deeper understanding of its unique molecular and cellular characteristics. This effort involves identifying novel therapeutic targets within the tumor and its microenvironment.

Pancreatic cancer is histologically characterized by its tumor microenvironment, which is marked by collagenous and dense desmoplasia/stroma, hypoxia, and significant infiltration of immunosuppressive cells, which collectively create a barrier to effective drug delivery and impede the efficacy of immune-based therapies.

The stromal reaction in pancreatic ductal adenocarcinoma is a hallmark of the disease, characterized by a complex interplay of diverse cellular and non-cellular components. This reaction is not merely a passive response to the growing tumor but rather an active contributor to the tumor’s progression and its resistance to therapy.

Among the cellular components, pancreatic stellate cells play an essential role. When activated in disease states such as inflammation or cancer, pancreatic stellate cells produce large amounts of extracellular matrix proteins. This production contributes to the fibrosis of chronic pancreatitis and desmoplasia of pancreatic cancer [12]. Endothelial cells are also important, as they form the blood vessels that supply the tumor with nutrients and oxygen, facilitating its growth and the potential for metastasis. The immune cells present within the stroma, including macrophages, lymphocytes, and neutrophils, often adopt an immunosuppressive phenotype. This adaptation not only aids in tumor evasion from immune surveillance but also creates a pro-tumorigenic environment. Additionally, neural elements within the stroma contribute to the pain commonly associated with pancreatic cancer and may also play a role in tumor growth and spread [13]. As the non-cellular components, the stroma is rich in various extracellular matrix components, such as collagens, which provide structural support to the tumor and contribute to its characteristic desmoplastic reaction [13]. Glycoproteins, fibronectin, hyaluronic acid, and proteoglycans are abundant and play roles in cell adhesion, migration, and the creation of a barrier to effective drug delivery. These components also contribute to the overall stiffness and hypoxic conditions within the tumor microenvironment.

Furthermore, the stromal reaction involves a multitude of growth factors, cytokines, and the serine protein, which is acidic and rich in cysteine [13]. These factors not only support tumor growth and survival but also contribute to the modulation of the immune response and the promotion of angiogenesis. Growth factors such as transforming growth factor-beta and vascular endothelial growth factor are particularly notable for their roles in promoting tumor growth and angiogenesis, respectively.

This complex interplay between the tumor cells and their microenvironment is a key factor in the refractory nature of these tumors. This unique and hostile microenvironment poses significant hurdles, not only reducing the efficacy of conventional chemotherapy but also impeding the success of immune therapies.

In recent years, many studies have revealed a complex bidirectional communication between pancreatic stellate cells and pancreatic cancer cells. This interaction is a critical driver of cancer cell proliferation, progression, and metastasis. It also influences the behavior of pancreatic stellate cells, including their proliferation, migration, and production of extracellular matrix components [12,14,15,16,17,18,19]. The HGF/c-MET (mesenchymal–epithelial transition factor) pathway is emerging as a key area of focus in understanding and potentially disrupting the intricate cellular interaction that results in pancreatic cancer’s aggressiveness and resistance to treatment.

## 2. HGF/c-MET Pathway

c-MET, belonging to the MET family of proteins, is a receptor tyrosine kinase that is expressed on the surfaces of a variety of epithelial cells. This includes cells found in pancreatic cancer and endothelial cells, making it a critical component in the study of these cell types. Its primary ligand, hepatocyte growth factor (HGF), is predominantly produced by pancreatic stellate cells, as evidenced by studies [20,21]. In addition to its presence in pancreatic stellate cells, HGF is synthesized by fibroblasts, mesenchymal cells, and smooth muscle cells. It is a member of the soluble cytokine family and is also categorized within the plasminogen-related growth factor family, denoting its diverse biological roles [22]. The interaction of HGF with c-MET activates numerous downstream signaling pathways, playing a pivotal role in a variety of crucial biological processes. The HGF/c-MET pathway is especially important in mediating vital processes such as embryogenesis, the healing of wounds, and the regeneration of tissue, including the development of muscle and nerve tissues. This pathway’s involvement in these processes highlights its critical role in human biology, contributing to various essential functions and developmental stages in humans [23,24,25]. Given its wide range of influence, the HGF/c-MET pathway has been the focus of numerous studies, aiming to reveal its complexities and the potential implications of its dysregulation in pathological conditions such as cancer, fibrosis, and other diseases. The intricate nature of this pathway underscores the need for comprehensive research to fully understand its mechanisms and potential therapeutic targets, especially in the context of diseases like pancreatic cancer, where it plays a crucial role.

The gene responsible for encoding MET, also known as c-MET, is comprised of a complex structure featuring 21 exons and 20 introns. This gene is responsible for the production of a specific protein that has a molecular weight of approximately 120 kilodaltons (kDa) [26]. The protein produced by the c-MET gene is notable for its structure: it forms a heterodimer, which is a compound formed from two different molecules. In this case, these molecules are the 50 kDa extracellular α chain and the 145 kDa transmembrane β chain, both of which play crucial roles in the protein’s function and interaction with its ligand. The extracellular α chain is the part of the protein that is exposed to the external environment of the cell, while the transmembrane β chain spans the cellular membrane [27].

One of the key functional domains of this protein is the SEMA domain, located within the transmembrane chain. This domain is of particular interest because it contains the specific binding site for HGF, a critical ligand for c-MET. The interaction between HGF and c-MET in this domain is essential for the activation of various downstream signaling pathways that play significant roles in cellular processes [27].

Furthermore, the *HGF* gene itself, which is situated on human chromosome 7, encodes a protein consisting of 728 amino acids. This genetic composition includes 18 exons and 17 introns, further illustrating the complexity of the gene’s structure [26]. The interplay between HGF and c-MET, regulated by their respective genetic features and protein structures, is fundamental to understanding their roles in cellular signaling and potential implications in various medical conditions.

The HGF protein is structured as a heterodimer consisting of two distinct chains, the α chain and the β chain, with molecular weights of 69 kDa and 34 kDa, respectively. The β chain is particularly notable for its serine protease analog domain. This domain plays a pivotal role, as it constitutes the binding site for the c-MET receptor, a critical interaction point in cellular signaling pathways [28]. The activation of HGF is an intricate process: it achieves its active form through a mechanism known as proteolytic cleavage. This process involves the enzymatic breaking of certain bonds within the protein, leading to structural changes that activate the protein. Once activated, HGF functions as the exclusive ligand for c-MET.

The binding of HGF to its c-MET receptor is not merely a static interaction but triggers a series of consequential events. This binding leads to the dimerization and phosphorylation of c-MET, a process that activates the receptor and its several associated signaling pathways. The pathways activated include MAPK/ERK (MET), STAT3, and PI3K. These pathways play crucial roles in regulating various cellular behaviors, particularly those relevant to cancer biology. They influence key processes such as the proliferation, invasion, and migration of cancer cells, all of which are fundamental aspects of cancer progression and metastasis (Figure 1) [29,30].

The HGF/c-MET pathway, pivotal in embryonic development, plays a vital role in various stages of cell growth and organ formation. This pathway’s influence extends to wound healing, where it aids in tissue regeneration and repair, which is crucial for recovery from injuries. Its dysregulation can result from gene amplifications, genetic mutations, protein overexpression, or a ligand-dependent autocrine or paracrine signaling loop [31,32].

## 3. HGF/c-MET Pathway in the Pancreatic Cancer

HGF is produced by pancreatic stellate cells, and its receptor, c-MET, is expressed in pancreatic cancer cells and endothelial cells [16,33].

### 3.1. HGF/c-MET and MAPK/ERK (Ras, Ras/Raf/MEK/ERK) Pathway

When HGF binds to the c-Met receptor, a series of conformational changes are initiated in c-MET. These changes are crucial as they activate the protein tyrosine kinase (PTK) domain within the cell [34]. This activation is a critical first step in a cascade of signaling events. The activated PTK domain undergoes autophosphorylation, a process where phosphate groups are added to the protein, altering its activity and function. This phosphorylation enables the PTK domain to interact with the SH2/SH3 domain of the adapter protein Grb2.

This interaction with Grb2 is significant as it leads to the recruitment and subsequent activation of guanine exchange factors (GEFs), including SOS. These GEFs are instrumental in the activation of Ras-GTP (guanosine triphosphate), a key molecule located at the cell membrane. The activation of Ras-GTP marks the beginning of a complex signaling cascade. Once activated, Ras stimulates a series of kinases, including Raf, MEK, and multiple members of the MAPK family, such as ERK, JRK, and p38. This signaling pathway can interfere with cell transformation, the cell cycle, proteins, and matrix degradation, promote cell migration, and contribute to tumor proliferation [35,36].

The *Ras* gene, which is a critical component of this pathway, exhibits a high mutation rate. In pancreatic cancer, mutations in the *Ras* gene are present in approximately 85% of cases, and in colon cancer, around 40% [37]. These mutations often involve point mutations and gene amplifications, leading to the continuous activation of the MAPK/ERK pathway. Such persistent activation can contribute to uncontrolled cell proliferation, the evasion of apoptosis, and increased tumor invasiveness, ultimately promoting tumor growth and metastasis.

### 3.2. HGF/c-MET and PI3K Pathway

The phosphorylated site on c-MET serves as a docking site for the phosphatidylinositol-3-kinase (PI3K)-p85 subunit. This interaction is critical in the activation of the PI3K/AKT signaling pathway, a key regulator in cellular processes. The binding of the p85 subunit of PI3K to the SH2/SH3 domain of phosphorylated c-MET leads to a series of biochemical reactions. One of the most important of these is the transformation of PIP2 (phosphatidylinositol-4,5-diphosphate) into PIP3 (phosphatidylinositol-3,4,5-triphosphate). This conversion is a vital step in the signaling cascade, as PIP3 serves as a secondary messenger in multiple intracellular pathways.

The phosphorylated receptor tyrosine kinases, including c-MET, play a key role in this process by binding to p85, which subsequently recruits the p85-p110 PI3K complex to the cell membranes, activating it. Once active, PI3K catalyzes the conversion of PIP2 to PIP3. This conversion is significant as this interaction with signaling proteins that have a PH domain, specifically Akt (protein kinase B) and phosphoinositide-dependent protein kinase-1, leads to the phosphorylation of Akt [37].

Once activated, Akt serves as a critical modulator in multiple cellular processes, particularly those related to cell survival and proliferation. It influences various downstream transcription factors, such as forkhead transcription factor-like 1, nuclear factor kappa-light-chain-enhancer of activated B cells, and B-cell lymphoma 2. Akt’s modulation of these factors leads to the inhibition of tumor suppressor gene expression and the phosphorylation of critical enzymes and regulators such as glycogen synthase kinase-3 and the mammalian target of rapamycin (mTOR). These actions collectively contribute to tumorigenesis, highlighting the significance of the PI3K/AKT/mTOR pathway in cancer development and progression [38,39]. Furthermore, this pathway’s impact extends to modulating the expression of critical factors in the tumor microenvironment, such as hypoxia-inducible factor-1 (HIF-1) and vascular endothelial growth factor. This modulation occurs through the activation of proteins like human double minute 2 [40]. Each of these steps and interactions underscores the complexity and importance of the PI3K/AKT/mTOR signaling pathway in the context of oncology, particularly in its role in promoting tumor growth and survival.

### 3.3. HGF/c-MET and the Hypoxia, Angiogenesis, Metastasis of Pancreatic Cancer

Pancreatic cancer is characterized by a highly hypoxic microenvironment, a condition in which there is a deficiency of oxygen within the tumor tissue. This hypoxic state is a critical factor in tumor progression and is known to activate hypoxia-inducible factor 1-alpha (HIF-1α). HIF-1α is a transcription factor that responds to low oxygen levels in the cellular environment [41,42]. Pancreatic cancer has a highly hypoxic microenvironment in which HIF-1α is activated, and it subsequently activates the MET in pancreatic cancer, promoting stromal–tumor communication and inducing neo-angiogenesis [43,44]. It is reported that the upregulation of c-MET is required in angiogenesis [45].

The critical role of the HGF/c-MET pathway in terms of its mitogenic and motogenic effects on cancer cells has been reported in several cancers, including pancreatic cancer [33,46,47,48]. Tumor expression and the phosphorylation of c-MET have been associated with poor survival and early distant metastases. Elevated serum HGF levels have been correlated with disease progression in pancreatic cancer patients [49,50,51,52]. Additionally, c-MET activation has been reported to increase resistance to chemotherapy, tumor cell motility, and the secretion of angiogenic factors in patients with pancreatic cancer [53,54,55].

### 3.4. HGF/c-MET and the Urokinase Plasminogen Activator (uPA) Positive Feed-Forward Loop

uPA is a serine protease that plays an important role in various physiological and pathological processes, including cancer progression. In the context of pancreatic cancer, uPA is particularly significant due to its essential function in activating the HGF/c-MET pathway. uPA achieves this by cleaving the inactive form of HGF, a process that activates HGF and enables it to bind to its receptor, c-MET [56]. Additionally, the binding of HGF to the c-MET receptor on cancer cells triggers a cascade of downstream signaling events that further induces the production of uPA. This creates a continuous positive feedback loop wherein the activation of the HGF/c-MET pathway enhances uPA production, which in turn further stimulates the pathway [56].

6-substituted hexamethylene amiloride derivate, a uPA inhibitor, was reported to significantly reduce metastasis in an orthotropic pancreatic cancer model [57].

## 4. Trametinib, the MEK Inhibitor

Trametinib is an orally administered selective inhibitor specifically targeting MEK1 (MAPK kinase 1) and MEK2 (MAPK kinase 2). With a molecular weight of 615.4 g/mol, trametinib stands out due to its targeted action on the MAPK/ERK pathway [58,59]. This drug has emerged as a pivotal therapeutic option targeting the MAPK/ERK pathway in various malignancies, including BRAF-mutated melanoma, non-small cell lung cancer and thyroid cancer. The structure and mechanism of trametinib allow it to effectively inhibit the MEK1 and MEK2 proteins, thereby impeding the downstream signaling that leads to cancer cell proliferation and survival (Figure 1).

The enhanced effectiveness of trametinib is observed when it is used in combination with other therapeutic agents, especially BRAF inhibitors. This synergistic effect allows for a more comprehensive approach to targeting cancer cells, particularly those with specific genetic mutations like BRAF mutations. Trametinib’s approval by the Food and Drug Administration (FDA) in 2013 marked a significant milestone in melanoma treatment [59]. Trametinib is currently approved for various cancers with BRAF mutations, such as non-small cell lung cancer and thyroid cancer, either as monotherapy or in combination with dabrafenib, a BRAF inhibitor, for improved therapeutic efficacy. Trametinib’s role in treating non-small cell lung cancer and thyroid cancer, among others, has been a subject of ongoing research and clinical trials. The ongoing development and clinical application of trametinib represent a significant advancement in cancer therapy, offering hope for improved outcomes in patients with these challenging malignancies.

Studies of other MET tyrosine kinase inhibitors, such as crizotinib, cabozantinib, and capamatinib, for pancreatic cancer have suggested that trametinib may be effective in treating pancreatic cancer. Additionally, Beta 1 integrin signaling has been reported to mediate resistance to MEK inhibition by trametinib in pancreatic ductal adenocarcinoma [60]. The most common adverse effects of trametinib include rash, dermatitis, diarrhea, and fatigue [61]. Trametinib exhibits a favorable pharmacokinetic profile, characterized by a prolonged half-life, low peak-to-trough ratio, and limited toxicity when compared to other agents in the same class [62,63].

## 5. Mechanisms of Trametinib in Pancreatic Cancer

Trametinib, a MEK inhibitor, has shown potential in targeting pancreatic cancer through multifaceted mechanisms. These mechanisms not only target the cancer cells directly but also modulate the tumor microenvironment and immune responses, creating a comprehensive antitumor strategy.

### 5.1. Inhibition of the HGF/c-MET and MAPK/ERK Pathways

A key aspect of trametinib’s action is its blockade of the MAPK/ERK pathway, which is a downstream pathway of the HGF/c-MET signaling. The MAPK/ERK pathway, frequently activated in cancers due to mutations in genes like KRAS, plays a vital role in cell proliferation and survival [37]. By blocking this pathway, trametinib induces G1 cell cycle arrest and accelerates apoptosis by interfering with the conformation of the activation loop sites of MEK1/2. This alteration prevents effective phosphorylation by RAF. Despite promoting the activation of upstream components of the pathway, trametinib ensures persistent inhibition of MEK1/2 phosphorylation and ERK1/2 activity. This characteristic leads to a more sustained and effective inhibition. A report indicated that trametinib can prevent the phosphorylation of MEK1 at Ser218 but cannot at Ser222, and this monophosphate form of MEK1 has severely limited kinase activity [61,64].

However, in G12D KRAS mutation and KRAS wild type, resistance to MEK inhibitor is induced by the elevation of Akt activity [65]. Rapid adaptive resistance to KRAS and MEK inhibitors, mediated by integrin-linked kinase, resulting in increased phosphorylation of the mTORC2 component Rictor and AKT, has also been reported [66]. Therefore, a combination treatment of KRAS-MEK inhibitor with Akt inhibitor or mTORC1/2 inhibitors has been suggested. The simultaneous targeting of KRAS-MEK and mTORC1/2 prevents the upregulation of ERK and AKT phosphorylation, leading to the inhibition of protein translation and cancer cell survival pathways. This combination of inhibition increases cell death and mitigates pAKT-driven adaptive resistance.

### 5.2. Alterations in the Tumor Microenvironment and Modulation of Immune Responses

As mentioned above, the pancreatic adenocarcinoma, especially the basal-like mesenchymal pancreatic ductal adenocarcinoma, presents a unique challenge in cancer therapy due to its distinct and unique immunosuppressive tumor microenvironment, exacerbated by allelic imbalances and high expression levels of oncogenic KRAS [11]. This results in resistance to traditional targeted therapies and immunotherapies, which necessitates an innovative approach.

The combination therapy of the MEK inhibitor trametinib and the multi-kinase inhibitor nintedanib has emerged as a promising solution, targeting the complexities of basal-like mesenchymal pancreatic ductal adenocarcinoma. By remodeling the tumor microenvironment, this combination therapy of trametinib and nintedanib enhances the effectiveness of immune checkpoint inhibitors in mesenchymal-type pancreatic ductal adenocarcinoma. It achieves a significant shift in immune responses: reducing naive-like CD8+ and CD4+ T cells, which are typically less effective against tumors, and increasing intra-tumoral infiltration of cytotoxic and effector T cells. The cytotoxic T cells and effector T cells are known to sensitize the mesenchymal pancreatic ductal adenocarcinoma to the Programmed Death-Ligand 1 immune checkpoint inhibition. Overall, this combinational therapy enhances the immune system’s capacity to target and destroy cancer cells, particularly in mesenchymal-type pancreatic cancer (Figure 2) [11].

Additionally, this combinational therapy induces alterations of the tumor microenvironment of pancreatic ductal adenocarcinoma. Specifically, it induces reprogramming in the subtypes of cancer-associated fibroblasts, leading to a decrease in myofibroblast-like cancer-associated fibroblasts and an increase in inflammatory cancer-associated fibroblasts. This reprogramming potentially hinders the tumor-supportive functions of cancer-associated fibroblasts and promotes antitumor responses. Additionally, trametinib reduces *Tgfb1* expression in myofibroblast-like cancer-associated fibroblasts, thereby contributing to antitumor response [11].

This strategic modulation of both the tumor microenvironment and the immune system not only addresses the inherent resistance of basal-like mesenchymal pancreatic ductal adenocarcinoma to conventional therapies but also opens new avenues for more effective treatment strategies, including targeted therapy and immunotherapies, potentially leading to improved patient outcomes in this particularly challenging cancer.

## 6. Clinical Trials of Trametinib in Pancreatic Cancer

The landscape of clinical trials about pancreatic cancer and trametinib, as summarized in Table 1, presents a diverse array of studies exploring the efficacy of the drug in various combinations and settings [67,68,69].

A phase I trial (NCT04303403) is currently investigating the safety, tolerability, and potential benefits of trametinib in combination with ruxolitinib for patients with RAS mutant colorectal cancer and pancreatic adenocarcinoma, aiming to uncover new treatment avenues for these cancers [70].

Concurrently, another Phase I trial (NCT05440942) is exploring the effects of a triple combination therapy involving MEK, STAT3, and PD-I inhibitors (trametinib, retifanlimab, and ruxolitinib) in patients with metastatic pancreatic adenocarcinoma, with the goal of reducing tumor size and advancing treatment options [71].

The THREAD trial (NCT03825289), another phase I study, is focused on the side effects and optimal dosing of hydroxychloroquine when administered with trametinib in pancreatic cancer patients, potentially offering insight into the synergistic effects of this combination [72].

A phase II study (NCT02428270) evaluated the antitumor effects of GSK2256098 and trametinib in advanced pancreatic cancer, providing crucial data on the clinical outcomes of this combination [67]. This clinical trial evaluated 11 patients who had progressed after first-line palliative chemotherapy, treating them with GSK2256098 and trametinib. The primary endpoint was clinical benefit, defined as a complete response, partial response, or stable disease for at least 24 weeks. In this study, 10 patients experienced progressive disease as their best tumor response, and one had stable disease for 4 months. The median progression-free survival was 1.6 months, and the median overall survival was 3.6 months. No treatment-related severe adverse events were observed. Although the drug combination was well-tolerated, it did not demonstrate significant activity in treating unselected advanced pancreatic ductal adenocarcinoma, as indicated by the limited clinical benefit and progression-free survival observed in the trial participants.

Another phase II trial (NCT02704156) [68] with an open-label, randomized control design completed its assessment. The researchers evaluated the clinical outcomes of stereotactic body radiotherapy combined with pembrolizumab and trametinib as a novel treatment option for patients with locally recurrent pancreatic cancer post-surgery. The study, which enrolled 170 patients with histologically confirmed pancreatic ductal adenocarcinoma characterized by mutant KRAS and positive immunohistochemical staining of programmed death-ligand 1, divided them into two groups: one receiving stereotactic body radiotherapy, pembrolizumab, and trametinib, and the other receiving stereotactic body radiotherapy and gemcitabine. The results indicated a median overall survival of 14.9 months in the stereotactic body radiotherapy, pembrolizumab, and trametinib group, compared to 12.8 months in the stereotactic body radiotherapy and gemcitabine group. The study concluded that the combination of stereotactic body radiotherapy with pembrolizumab and trametinib could be an effective treatment for this patient population, warranting further investigation in phase 3 trials. 

Additionally, a multi-center single-arm phase II trial (PaTcH, NCT05518110) [73] is underway, aiming to explore primary and emerging resistance mechanisms in metastatic refractory pancreatic cancer treated with trametinib and hydroxychloroquine, an important step towards understanding and overcoming drug resistance.

Lastly, another multi-center randomized, double-blind, placebo-controlled phase II trial (NCT01231581) compared clinical outcomes of trametinib and gemcitabine combination therapy versus placebo and gemcitabine, offering valuable insights into the effectiveness of trametinib in combination treatments. In this study, 160 patients with untreated metastatic pancreatic ductal adenocarcinoma were equally divided into two groups: one receiving trametinib plus gemcitabine and the other gemcitabine with a placebo, with each group consisting of approximately 80 patients. The study reported similar median progression-free survival for both groups, with the trametinib plus gemcitabine group showing a duration of 16 weeks compared to 15 weeks for the gemcitabine plus placebo group. In terms of median overall survival, it was observed to be 8.4 months for patients treated with trametinib and gemcitabine and 6.7 months for those receiving gemcitabine with a placebo. The study concluded that the addition of trametinib to gemcitabine did not significantly improve overall survival, progression-free survival, overall response rate, or duration of response in patients with previously untreated metastatic pancreatic cancer [69].

## 7. Conclusions

Pancreatic cancer is characterized as an aggressive malignancy that poses challenges in early detection, compounded by its unique tumor microenvironment of desmoplasia. This distinctive microenvironment not only reduces the effectiveness of conventional chemotherapy but also hampers the success of immune therapies. Trametinib, an oral MEK1 and MEK2 inhibitor, demonstrates inhibitory effects on cancer cell mitosis, angiogenesis, and metastasis while modulating the tumor microenvironment. The multifaceted actions of trametinib highlight its potential as a novel therapeutic option for pancreatic cancer. Ongoing clinical trials are actively investigating its efficacy and safety, both as a standalone treatment and in combination therapy. These studies are crucial in defining trametinib’s role in the evolving landscape of pancreatic cancer treatment.

## Figures and Tables

**Figure 1 cancers-16-01056-f001:**
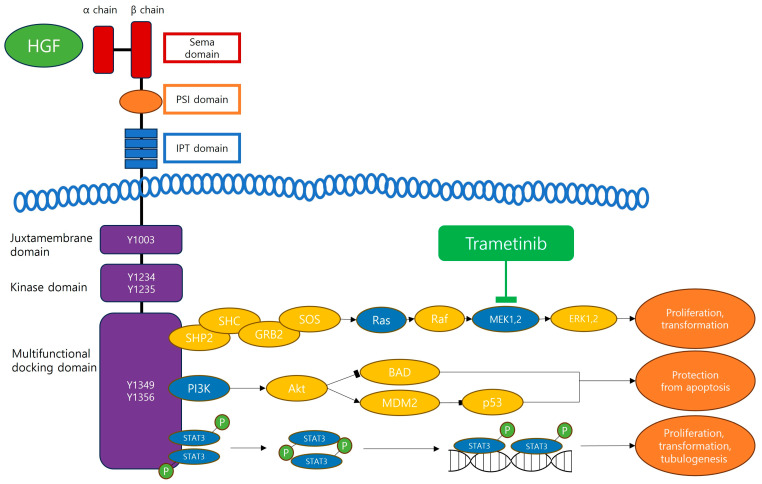
Here, the major downstream signaling pathways of HGF/c-MET and targets of trametinib are shown. MET triggers several downstream signaling pathways: the Ras pathway, the PI3K–Akt pathway, and the STAT3 pathway. Trametinib targets the Ras (Ras/Raf/MEK/ERK, MAPK) pathway in various malignancies. HGF, hepatocyte growth factor; SHP2, SH2 domain-containing tyrosine phosphatase 2; SHC, SH2 domain-containing transforming protein; GRB2, growth factor receptor-bound protein 2; SOS, son of sevenless; MEK, mitogen-activated protein kinase kinase; ERK, extracellular signal-regulated kinases; PI3K, Phosphoinositide 3-kinase; Akt, protein kinase B; BAD, BCL-2 antagonist of cell death; MDM2, murine double minute 2; STAT3, Signal transducer and activator of transcription 3.

**Figure 2 cancers-16-01056-f002:**
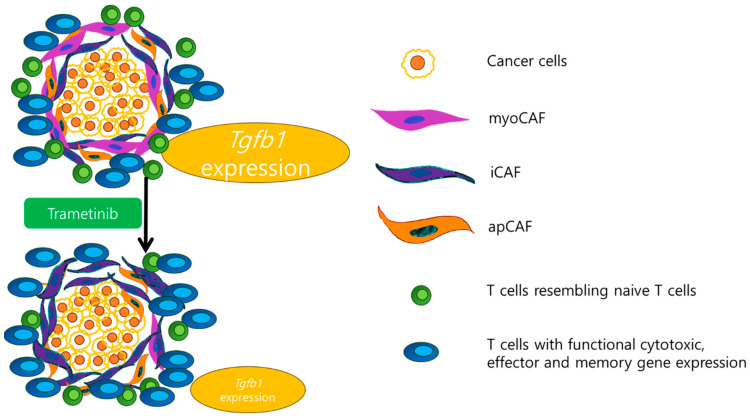
The figure shows tumor microenvironment remodeling by trametinib in pancreatic cancer. Trametinib induces significant alterations in the tumor microenvironment of pancreatic cancer. 1. Fibroblast Changes: (1) Decrease in myofibroblast-like cancer-associated fibroblasts (myoCAF). (2) Increase in inflammatory cancer-associated fibroblasts (iCAF). 2. *Tgfb1* Expression: (1) Reduces *Tgfb1* expression in myoCAF. 3. T Cell Responses: (1) Decrease in CD8+ and CD4+ T cells with gene expression resembling naive T cells. (2) Increase in T cells exhibiting functional cytotoxic, effector, and memory gene expression in mesenchymal pancreatic cancer.

**Table 1 cancers-16-01056-t001:** Clinical trials of trametinib in pancreatic cancer.

ClinicalTrials.gov ID	Study Design	Intervention	Diagnosis and Stage	Number of Patients Treated with Trametinib	Median Progression Free Survival (95% Confidence Interval)	Median Overall Survival (95% Confidence Interval)	Publication Year	PMID
NCT04303403	Phase Ib trial	Combination treatment of trametinib and ruxolitinib	Advanced RAS mutant colorectal cancer and pancreatic adenocarcinoma	NA	NA	NA		NA
NCT05440942	Phase I trial	Combination treatment of trametinib, retifanlimab, and ruxolitinib	Metastatic pancreatic ductal adenocarcinoma	NA	NA	NA		NA
NCT03825289	Phase I trial	Combination treatment of trametininb and hydroxychloroquine	Metastatic or locally advanced, unresectable pancreatic carcinoma	NA	NA	NA		NA
NCT02428270	Phase II trial of investigational drugs GSK2256098 and trametinib	GSK2256098 and Trametinib	Advanced PDAC patients whose disease progressed after first-line palliative chemotherapy	16	1.6 months (1.5–1.8 months)	3.6 months (2.7 months-not reached)	2022	36636049
NCT02704156	Phase II trial with an open-label, randomized control design	Stereotactic body radiation therapy plus pembrolizumab and trametinib	Locally recurrent pancreatic cancer after surgical resection	85	8.2 months (6.9–9.5 months)	14.9 months (12.7–17.1 months)	2022	35240087
NCT05518110	Phase II trial	Combination treatment of trametinib and hydroxychloroquine	Metastatic pancreatic cancer which has previously progressed on at least one line of systemic therapy	NA	NA	NA		NA
NCT01231581	Phase II trial with randomized, double-blind, placebo-controlled design	Combination treatment of trametinib and gemcitabine	Untreated metastatic adenocarcinoma of the pancreas	80	8.4 months (7.8–9.9 months)	16.1 weeks (14.0–23.4)	2014	24915778

NA, not available.

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
