# Peer review of "Pancreatic Cancer Treatment Targeting the HGF/c-MET Pathway: The MEK Inhibitor Trametinib"

_cancers, 2024, doi:10.3390/cancers16051056_

Round 1
Reviewer 1 Report
Comments and Suggestions for Authors
The manuscript is an interesting review on the effect of the MEK inhibitor, Trametinib on HGF/c-MET pathway.
This review can be accepted for publication in “Cancers” after minor revision.
In p. 4 the authors could add the following review in the references: 10.1038/s41388-021-01863-w (review 2021, HGF/c-MET pathway in cancer).
P.6 “In pancreatic cancer, HGF is predominantly secreted by pancreatic stellate cells, while c-MET is expressed by endothelial cells and cancer cells.[16, 47]” This sentence could be deleted since it is a repetition of the sentence in section 3. (p. 5)
P. 8 “combinatorial therapy” should be replaced by “combination therapy” Moreover, n in the sentence “The combinational therapy of n the MEK inhibitor trametinib” must be deleted.
P. 9: (Clinical Trials of Trametinib) References at the 2nd to 4th paragraphs are missing.
The references should be described according to the journal format.
Author Response
Reviewer 1
The manuscript is an interesting review on the effect of the MEK inhibitor, Trametinib on HGF/c-MET pathway.
This review can be accepted for publication in “Cancers” after minor revision.
In p. 4 the authors could add the following review in the references: 10.1038/s41388-021-01863-w (review 2021, HGF/c-MET pathway in cancer).
Response: Many thanks for your kind suggestion. We added the reference, “J. Fu et al., "HGF/c-MET pathway in cancer: from molecular characterization to clinical evidence," Oncogene, vol. 40, no. 28, pp. 4625-4651, Jul 2021, doi: 10.1038/s41388-021-01863-w.”
P.6 “In pancreatic cancer, HGF is predominantly secreted by pancreatic stellate cells, while c-MET is expressed by endothelial cells and cancer cells.[16, 47]” This sentence could be deleted since it is a repetition of the sentence in section 3. (p. 5)
Response: Thanks for your kind comment. The sentence was deleted according to your comment.
- 8 “combinatorial therapy” should be replaced by “combination therapy” Moreover, n in the sentence “The combinational therapy of n the MEK inhibitor trametinib” must be deleted.
Response: Thanks for your kind comments. The sentence was revised according to your comments.
- 9: (Clinical Trials of Trametinib) References at the 2ndto 4thparagraphs are missing.
The references should be described according to the journal format.
Response: Many thanks for your important comments. The references were added.
Reviewer 2 Report
Comments and Suggestions for Authors
This is a well written review article on the role of MEK-inhibition in pancreatic cancer. The manuscript needs some linguistic improvement. The structure of the manuscript is clear and the information regarding the background is provided in a comprehensive manner.
We would suggest that more information on resistance mechanisms during MEK-inhibition is provided and to include the describe of potential predictive biomarkers for MEK-inibitor treatment in pancreatic cancer and some more information on the the mechanisms of resistance and how to overcome this problem (for example Brauswetter D, Gurbi B, Varga A, Varkondi E, Schwab R, Banhegyi G, et al. Molecular subtype specific efficacy of MEK inhibitors in pancreatic cancers. PLoS One. 2017;12(9):e0185687; Brown WS, McDonald PC, Nemirovsky O, Awrey S, Chafe SC, Schaeffer DF, et al. Overcoming Adaptive Resistance to KRAS and MEK Inhibitors by Co-targeting mTORC1/2 Complexes in Pancreatic Cancer. Cell Rep Med. 2020;1(8):100131 and others).
Comments on the Quality of English LanguageSome improvement is needed.
Author Response
Reviewer 2
This is a well written review article on the role of MEK-inhibition in pancreatic cancer. The manuscript needs some linguistic improvement. The structure of the manuscript is clear and the information regarding the background is provided in a comprehensive manner.
We would suggest that more information on resistance mechanisms during MEK-inhibition is provided and to include the describe of potential predictive biomarkers for MEK-inhibitor treatment in pancreatic cancer and some more information on the the mechanisms of resistance and how to overcome this problem (for example Brauswetter D, Gurbi B, Varga A, Varkondi E, Schwab R, Banhegyi G, et al. Molecular subtype specific efficacy of MEK inhibitors in pancreatic cancers. PLoS One. 2017;12(9):e0185687; Brown WS, McDonald PC, Nemirovsky O, Awrey S, Chafe SC, Schaeffer DF, et al. Overcoming Adaptive Resistance to KRAS and MEK Inhibitors by Co-targeting mTORC1/2 Complexes in Pancreatic Cancer. Cell Rep Med. 2020;1(8):100131 and others).
Response: Thank you very much for your kind suggestion. We described this as follows:
“However, in G12D KRAS mutation and KRAS wild type, resistance to MEK inhibitor is induced by the elevation of Akt activity.[65] Rapid adaptive resistance to KRAS and MEK inhibitors, mediated by integrin-linked kinase, resulting in increased phosphorylation of the mTORC2 component Rictor and AKT, also has been reported.[66] Therefore, combination treatment of KRAS-MEK inhibitor with Akt inhibitor or mTORC1/2 inhibitors has been suggested. Simultaneous targeting of KRAS-MEK and mTORC1/2 prevents the upregulation of ERK and AKT phosphorylation, leading to the inhibition of protein translation and cancer cell survival pathways. This combination of inhibition increases cell death and mitigates pAKT-driven adaptive resistance.
Round 2
Reviewer 2 Report
Comments and Suggestions for Authors
With the changes made, the manuscript should now be fine for publication.
Comments on the Quality of English LanguageMinor editing is needed.